# Low n-6/n-3 Gestation and Lactation Diets Influence Early Performance, Muscle and Adipose Polyunsaturated Fatty Acid Content and Deposition, and Relative Abundance of Proteins in Suckling Piglets

**DOI:** 10.3390/molecules27092925

**Published:** 2022-05-04

**Authors:** Yron Joseph Yabut Manaig, Silvia Sandrini, Sara Panseri, Gabriella Tedeschi, Josep M. Folch, Armand Sánchez, Giovanni Savoini, Alessandro Agazzi

**Affiliations:** 1Departament de Ciència Animal i dels Aliments, Universitat Autònoma de Barcelona, 08193 Bellaterra, Barcelona, Spain; josepmaria.folch@uab.cat (J.M.F.); armand.sanchez@uab.cat (A.S.); 2Plant and Animal Genomics, Centre for Research in Agricultural Genomics (CRAG), CSIC-IRTA-UAB-UB Consortium, 08193 Bellaterra, Barcelona, Spain; 3Department of Veterinary Medicine and Animal Sciences, Università degli Studi di Milano, 26900 Lodi, Italy; silvia.sandrini@unimi.it (S.S.); sara.panseri@unimi.it (S.P.); giovanni.savoini@unimi.it (G.S.); alessandro.agazzi@unimi.it (A.A.); 4CRC “Innovation for Well-Being and Environment” (I-WE), Università degli Studi di Milano, 20122 Milano, Italy; gabriella.tedeschi@unimi.it

**Keywords:** piglets, longissimus dorsi, adipose tissue, PUFA, proteomics, omega-6, omega-3, inflammation, fat deposition

## Abstract

Elevated omega-6 (n-6) and omega-3 (n-3) polyunsaturated fatty acids (PUFAs) ratios in swine diets can potentially impose a higher risk of inflammatory and metabolic diseases in swine. A low ratio between the two omega PUFAs has beneficial effects on sows’ and piglets’ production performance and immunity status. At present, there are few studies on how sow nutrition directly affects the protein and fat deposition in suckling piglets. Two groups of sows were fed diets with high or low n-6/n-3 polyunsaturated ratios of 13:1 (SOY) and 4:1 (LIN), respectively, during gestation and lactation. Longissimus dorsi muscle and adipose tissue from newborn piglets, nourished only with sow’s milk, were subjected to fatty acid profiling by gas chromatography–mass spectrometry (GC-MS) and to proteomics assays based on nano-liquid chromatography coupled to high-resolution tandem mass spectrometry (nLC-HRMS). Fatty acid profiles on both muscle and adipose tissues resembled the magnitude of the differences between fatty acid across diets. Proteomic analysis revealed overabundance of 4 muscle and 11 adipose tissue proteins in SOY compared to LIN in both piglet tissues. The detected overabundance of haptoglobin, an acute-phase protein, and the stimulation of protein-coding genes and proteins related to the innate immune response and acute inflammatory response could be associated with the pro-inflammatory role of n-6 PUFAs.

## 1. Introduction

Commercial swine feed is mostly cereal- and soybean meal-based, accompanied by plant and plant oil sources largely comprising omega-6 (n-6) polyunsaturated fatty acids (PUFAs). Consequently, this makes the amount of n-6 in feed mixes about 10 times higher than omega-3 (n-3) PUFAs [1]. Mammals, for example pigs, cannot incorporate double bonds in fatty acids beyond carbon 9 and 10 as they lack the required desaturase enzyme for their de novo synthesis; hence, essential fatty acids such as n-6 and n-3 should be supplemented in their diet [2]. As the trend for n-6 PUFAs in swine diets is growing, the continuous ingestion of n-6-enriched diets can cause an imbalance between the two PUFAs. As precursors of eicosanoids resulting in antagonistic inflammatory functions (anti- and pro-inflammatory, respectively), n-3 and n-6 PUFAs may induce substrate competition that can potentially affect metabolic health and inflammatory modulation [3,4]. Eicosanoids are bioactive lipid mediators synthesized through PUFA oxygenation, the majority of which are derived from n-6 PUFAs. Eicosanoid signaling has been proposed as a primary pro-inflammatory component of innate immunity and can control immune system activity [5,6].

Low dietary ratio between n-6 and n-3 (i.e., 4:1) can improve weaning weight, survival, weight gain, and influence the total n-3 polyunsaturated fatty acids found in colostrum and milk [7]. Previous human and pig studies on n-3 supplementation have shown reduced risk of cardiovascular diseases, obesity, and metabolic syndrome and diseases, and even beneficial effects on placental metabolism, inflammatory status, and lipid transfer [8,9,10,11,12]. Sow diet influences the accumulation of fatty acids by piglets through placental lipid transfer. Diets during gestation directly affect sow’s milk and piglet’s plasma fatty acid composition. This also indicates how maternal adipose tissue acts as the first depot of dietary fatty acids, which are then mobilized around farrowing and transferred to piglets [13]. In addition, evidence has shown how maternal dietary composition could possibly influence the availability and similarity of fatty acids in human placenta. Placental expression of G protein-coupled receptor 120 (GPR120, a decosahexaenoic acid receptor) has been correlated with adipocyte differentiation in neonatal fat [11,14]. A study on rats has also shown on how the inclusion of n-3 PUFAs increased the growth of the fetus and placenta and reduced the oxidative degradation of lipids by increasing the expression of antioxidant enzymes in placental zones [15]. Proteomic approaches are now being used to assess intramuscular fat (IMF) deposition and pork quality. Pigs with high IMF revealed upregulated phosphatidylinositols and phosphatidylserines, whereas potential biomarkers such as α-actin, myosin-1, and myosin-4, were correlated with the degradation of myofibrillar proteins, indicating proteolysis and meat adhesiveness [16,17]. Unfortunately, none of these studies considered any proteomic approaches or elucidated the direct effects of gestating and lactating sow nutrition on the proteome of pre-weaning piglets. To our best knowledge, this work is one of the first studies to apply a proteomic approach in pre-weaned piglets fed with different n-6/n-3 ratio diets. The study was conducted to determine on how the milk of the sow, when fed with high or low n-6/n-3 PUFA ratio diets, directly affects and influences the fat deposition in muscle and in adipose tissues of pre-weaned piglets and the abundance of proteins, protein-coding genes, and their related ontologies and biological pathways.

## 2. Results

### 2.1. Sow Reproductive Performance

The overall reproductive performance of eight sows, four per treatment group, is summarized in Table 1. Dietary treatments had no effect (*p* > 0.05) on sow body weight, weight gain during gestation, or lactation periods. In addition to, SOY and LIN did not affect the total number of piglets born (LSM ± SEM: SOY 15.5 ± 1.31 piglets; LIN 14.0 ± 1.31 piglets; *p* > 0.05), but subsequently, SOY increased the number of dead-born piglets compared to LIN (SOY 2.50 ± 0.41 piglets; LIN 0.50 ± 0.41 piglets; *p* = 0.01). The total piglets weaned per sow in the SOY group (12.00 ± 1.34 piglets) and the LIN group (12.50 ± 1.34 piglets) were similar (*p* > 0.05). Furthermore, no significant differences in pre-weaning mortality were found between SOY and LIN (22.50% ± 0.06% vs. 11.0% ± 0.06%, respectively; *p* > 0.05).

### 2.2. Growth Performance of Litter and Pre-Weaning Piglets

Piglet growth performance is reported in Table 2. Both litter weight (LW) and litter weight gain (LWG) were not influenced (*p* > 0.05) by the dietary treatments during lactation (data not shown). Higher individual body weight was detected in LIN piglets on day (d) 15 of lactation (SOY 4.75 ± 0.16 kg; LIN 5.30 ± 0.16 kg; *p* = 0.02), but no differences between LIN and SOY were observed at birth or weaning. As for piglet body weight gain (PBWG), improved performance was found in the first two weeks of life in LIN (SOY 3.19 ± 0.14 kg; LIN 3.85 ± 0.14 kg; *p* = 0.0006), while higher PBWG was reported in SOY from d 15 up to weaning (SOY 2.72 ± 0.13 kg; LIN 2.22 ± 0.13 kg; *p* = 0.01). Over all the lactation period, similar PBWG was found in LIN and SOY groups of piglets (SOY 5.95 ± 0.20 kg; LIN 6.04 ± 0.20 kg; *p* = 0.75). The average daily gain (ADG) between SOY and LIN litters did not differ from birth to weaning, whereas LIN diet increased the ADG of individual piglets within its litter group (SOY 0.21 ± 0.01 kg; LIN 0.26 ± 0.01 kg; *p* = 0.0006). No differences were found in the ADG between SOY and LIN piglets from d 15 to weaning and in the overall weaning period.

### 2.3. Fatty Acid Composition of Muscle and Adipose Tissue Samples

A lower n-6/n-3 ratio in sow diets was reflected in a lower n-6/n-3 muscle (SOY 16.45 ± 0.40, LIN 9.72 ± 0.40; *p* < 0.0001) and adipose tissue (SOY 14.70 ± 0.17; LIN 7.47 ± 0.17; *p* < 0.0001) content in piglets, mainly due to the increased relative proportion of n-3 PUFAs in both tissues (SOY 1.26 ± 0.05; LIN 2.12 ± 0.05; *p* < 0.0001 and SOY 1.07 ± 0.02; LIN 1.91 ± 0.02; *p* < 0.0001; respectively) (Figure 1b). The administration of a diet with a low n-6/n-3 ratio also led to a decrease in the n-6 content in the piglet’s adipose tissue, but was not observed in muscle tissue. In the present trial, the total saturated fatty acids (SFA), monounsaturated fatty acids (MUFA), and PUFAs content of neither muscle nor adipose tissues were influenced by the adapted dietary treatment of the sow (Figure 1a), but specific differences were found in these tissues for the major n-6 PUFAs (Figure 1c) and major n-3 PUFAs (Figure 1d). Supplementing gestating and lactating sows with a n-6/n-3 ratio in the diet did not affect the relative proportion of the major n-6 PUFAs in the muscle of the piglet, but led to lower proportions of linoleic acid (LA), gamma γ-linoleic acid (GLA), and arachidonic acid (AA) in piglet adipose tissue. The low rate of n-6/n-3 in the maternal diet caused a generalized increase in all the major n-3 PUFAs in both the muscle and adipose tissue of the piglets.

### 2.4. Proteins Identification and Relative Abundance

#### 2.4.1. From Muscle Tissues

A total of 339 proteins (Appendix A) from the longissimus dorsi muscle were identified by Proteome Discoverer 2.5 software [18]. The identified muscle proteins were reported with their designated UNIPROT Accession number [19]. Abundance ratio comparison between dietary treatments SOY and LIN demonstrated four differentially expressed proteins (P_adj_ < 0.05), namely interferon-induced GTP-binding protein Mx2, prophenin and tritrpticin precursor, phosphoglycerate kinase 2, and haptoglobin. Moreover, three proteins—such as myoglobin, liver carboxylesterase, and protegrin-3—tended (P_adj_ < 0.10) to be differentially expressed (Table 3). 

#### 2.4.2. From Adipose Tissue

Proteome Discoverer 2.5 software identified a total of 389 proteins (Appendix A) from adipose tissue. Applying the same abundance ratio comparison (SOY vs. LIN), we obtained a total of 11 differentially expressed proteins (P_adj_ < 0.05)—60S ribosomal protein L29, myozenin-1, myosin light chain 4, myosin-4, haptoglobin, protegrin-2, liver carboxylesterase, phosphoglycerate kinase 2, histone H1t, glutathione S-transferase alpha M14, and desmoglein-1. In addition, we also included those proteins that tendend (P_adj_ < 0.10) to be overabundant in SOY over LIN, namely 60S ribosomal protein L35, metallothionein-1D, sarcoplasmic/endoplasmic reticulum calcium ATPase 3, and inter-alpha-trypsin inhibitor heavy chain H4. Their accession number, abundance ratio, and the genes encoding the aforementioned proteins are shown in Table 4.

### 2.5. Gene Ontologies and Biological Pathways

Due to the limited number of differentially expressed proteins (*p* < 0.05), genes encoding the abovementioned proteins were further investigated individually to generate ontologies and enrichment analysis. According to the UniProt database, genes found in muscle samples were related to the innate immune response (HP, MX2, P51524), glycolysis and gluconeogenesis (PGK2), and the acute inflammatory response (HP). Furthermore, genes found in adipose tissue samples exhibited relationships to GO pathways associated with the innate immune system (DSG1, NPG2), atherosclerosis (GSTA2), calcium ion binding and motor activity (MYH4, MYL4), cellular processes (GSTA2, RPL29, MYOZ1, HP), fatty acyl and cholesterol ester metabolism (CES1), glycolysis and gluconeogenesis (PGK2), and the acute inflammatory response (HP).

## 3. Discussion

The direct effect of high (13:1) and low (4:1) n-6/n-3 PUFA ratios in the sow’s diet was investigated on the reproductive performance of sows, growth performance of pre-weaning piglets and the fat deposition in muscle and in adipose tissues, given that the piglets were only nourished with sow’s milk. Moreover, changes in protein abundance, protein-coding genes, and their ontologies were further examined using proteomics data obtained from piglet tissues.

In our trial, LIN did not increase the total number of piglets born or weaned, but significantly decreased the number of dead-born piglets. Different sow studies found that n-3 PUFAs were able to decrease piglet mortality and improve pre- and post-weaning growth rates. Moreover, n-3 supplementation in the maternal diet was reviewed, along with how this benefits sow reproduction and piglet performance [20].

Piglet birth weight did not differ among dietary treatments, whereas an increase on piglet BW at d 15 was observed in the LIN group. Increased n-3 in gestational diets tends to cause increased piglet birth weight [21]. A meta-analysis in human studies also substantiated this claim, whereby n-3 PUFAs addition alone improved infant birth weight and was thus correlated to increased concentrations of n-3 PUFAs DHA and EPA [22]. Piglets from the SOY group had increased weight gain from d 15 to weaning. This can be attributed to the high AA concentration from the maternal diet, as major n-6 PUFAs (i.e., AA) have shown a positive correlation to neonatal growth. The association of AA content in piglet tissues and low birth weights was linked to the supplementation of n-3 PUFA in sow diets. The variability of the results on piglet growth performance may be related to substrate competition that can occur between the two PUFAs since n-3 and n-6 both serve as eicosanoid precursors and their inflammatory functions, synthesis, and oxidation pathways contrast each other [20,21,23]. We observed that litter weight and litter weight gain did not differ across dietary treatments and across defined subperiods. Among the sow reproductive studies that were reviewed, litter weight at birth did not significantly change upon the addition of n-3 PUFAs [20]. Although piglet weight and weight gain varied in between n-6/n-3 PUFA ratios, weaning weight and overall weight gain did not differ among dietary treatments.

Fatty acid profile on muscle and adipose tissues showed similarity to the dietary treatments and the same trend for total n-3, n-6, and n-6/n-3 ratios. From our results, sows fed SOY diets (n-6/n-3 ratio of 13:1) produced piglet with muscle and adipose tissue n-6/n-3 ratio of 16.45:1 and 14.70:1, respectively, while piglets from sows fed LIN (n-6/n-3 ratio of 4:1) had an n-6/n-3 ratio of 9.72:1 for muscle tissue and 7.47:1 for adipose tissue. Dietary ALA intake is shown to increase the DHA levels in piglet brain and EPA concentrations in the liver and in the blood of growing pigs [24,25]. This could explain the higher level of EPA and DHA levels with LIN compared to SOY, considering the high level of ALA in linseed oil. Furthermore, the biosynthesis of DHA in vertebrates is now widely accepted. It follows a pathway called the *Sprecher* pathway, in which it is achieved by two consecutive elongations from EPA to produce tetracosapentaenoic acid (TPA, 24:5n−3), which then undergoes a ∆6 desaturation to tetracosahexaenoic acid (THA, 24:6n−3), the latter being β-oxidized to DHA in peroxisomes [26]. Thus, n-3 and n-6 PUFAs act as both a precursor and inhibitor, as they share the same enzymes for the synthesis of long-chain PUFAs. Following the magnitude of the difference between the omega ratio content of the diets and how this was reflected in piglet tissue, a clearly deposition pattern of fatty acid into the animal was shown. Dietary fat consumed by the pigs directly affect the fatty acid composition of the carcass. It is generally dependent and mimics the fatty acid composition of the diet [27,28]. During digestion, dietary fatty acids are minimally hydrogenated or remain unchanged. If the pigs are fed above the maintenance requirement, the efficiency of dietary fat utilization is 90% [29]. The same n-6/n-3 ratios were used in sow gestation and lactation diets as were implemented in this study and colostrum and milk samples were collected at d 7 and at the end of lactation period [7]. It was noted how the low n-6/n-3 ratio increased the level of n-3 PUFAs, especially ALA, and decreased n-6/n-3 ratios in colostrum and milk samples. They also concluded that there was interaction between sampling point and sow diet. These further supported how the FA profiles of colostrum and milk, given that the piglets used in this trial were only fed with sow’s milk, were mirrored and then incorporated into the muscle and subcutaneous fat of suckling pigs. In response to dietary fat ingestion, de novo synthesis of fatty acids is inhibited in favor of the direct deposition of fatty acids in adipose tissue, in which 31–40% of dietary fat is transferred to carcass lipids and to the intramuscular fat of muscle tissues (i.e., longissimus dorsi, semimembranosus, biceps femoris, diaphragm, masseter), regardless of fat source [30,31,32,33].

The relative proportion of fatty acids found in muscles was significantly higher in percentages than in adipose tissue, whereas the differences among fatty acids within adipose tissues in two dietary treatments were more defined. The accumulation of body fat occurs through cell hyperplasia (increase in cell number) and hypertrophy (increase in cell size) after birth. During the early life stages, adipocyte hyperplasic development is favored; developed lipocytes become the first fat depots, which are usually found in perirenal, mesenteric, and intermuscular fat. As the animal grows, they continue to accumulate and manufacture more fat cells in subcutaneous and intramuscular deposits [34]. Hence, from a developmental point of view, intramuscular fat (IMF) is the last adipocytes depot in animals. Fat usually follows an order of deposition from perirenal fat, intermuscular fat, subcutaneous fat, and finally, through to intramuscular fat. We may hypothesize that the adipocytes found in subcutaneous fat have already reached their maximum storage capacity which, in turn, deposited all the excess available fat in the muscle tissue, thus increasing the amount of fat stored in the longissimus dorsi muscle. Furthermore, the IMF accumulation in muscle are dependent on the number and metabolic activity of adipocytes, growth rate of muscle tissues, and the metabolic activity of organs, such as the liver [35]. The longissimus dorsi muscle normally contains 1% of the total lipid found in pigs [36]. The age (d 26) of the piglets upon the collection of biological samples should also be taken into consideration, since subcutaneous carcass fat and total muscle lipid composition predominantly increase with the maturation of growing pigs [37]. Our results have also demonstrated how the relative proportion of n-6 PUFAs among dietary treatments resulted in more significant variations in adipose tissue than in muscle tissue, with the opposite for n-3 PUFAs. This may be due to high PUFA concentration of membrane lipids in the IMF, making them less vulnerable to nutritional modification [38].

The overabundance of muscle and adipose proteins found with SOY diets primarily indicated associations with pathways related to immune response, fatty acid metabolism, glycolysis and gluconeogenesis, and the inflammatory response. Due to contrasting function between n-6 and n-3 PUFAs, an increased supply of SOY in diets may trigger the pro-inflammatory function and effects of n-6 PUFA. The n-6/n-3 ratios of 9:1 and 13:1 have reportedly increased the immunoglobulin concentration in sow colostrum, although only 9:1 had effect on milk and in piglet plasma at d 21 of lactation. Although the mode of action of n-6 and n-3 PUFAs on immunoglobulins has not yet been elucidated, PUFAs are known to be involved in the production of white blood cell-derived cytokine called interleukins [39]. Eicosanoids are bioactive lipid mediators synthesized through polyunsaturated fatty acid oxygenation. The majority of eicosanoids are produced from n-6 PUFA AA, while some arise from processing of n-3 PUFAs such as EPA or DHA. Eicosanoid signaling, as with cytokine signaling and inflammasome formation, has been associated as a primary pro-inflammatory component of innate immunity [5,6]. Although our results did not show any significant difference between the distribution of AA in muscle tissues, EPA was significantly and proportionally higher in adipose tissue. These two PUFAs act as competing substrates to cyclooxygenase (COX) and lipoxygenase (LOX) enzymes for eicosanoid synthesis [40]. The production of pro-inflammatory cytokines was further correlated to EPA/DHA:AA ratio in the membrane phospholipids of mononuclear cells, in human alveolar cells, and in smooth muscle cells [41,42,43]. Other pathways involving the overabundant proteins in adipose tissue are mostly related to fatty acid metabolism and processes and can be related to the addition of PUFAs in the diet.

Likewise, a total of four common overabundant proteins were found in both muscle and adipose tissue samples. These are haptoglobin, phosphoglycerate kinase 2, liver carboxylesterase, and porcine antimicrobial peptides protegrin (i.e., protegrin-2 and protegrin-3). The increased amount of haptoglobin in both muscle and adipose tissue could be related to the pro-inflammatory role of n-6 PUFAs. Animals that are subjected to health or stress-related challenges activate both their innate and acquired immune systems. The innate immune system involves the host defense mechanism, the production of antibodies, and leukocyte activity, whereas the acute-phase response is the biological reaction to infection, inflammation, or trauma. One of the pathophysiological responses is the plasma protein production, mainly in the liver, known as acute-phase proteins [44]. Haptoglobin is considered as one of the main acute-phase proteins in pigs and used as a diagnostic tool to assess diseases, health status, and production performance. As it is mostly synthesized in the liver and dependent on the synthesis of interleukin 6 (IL-6), it also protects the host against the dangers of acute-phase reactions [45,46,47]. In pigs, weaning is considered as a stressful event that can impact growth rate, gut health, immune functions, and nervous system functions that extends until maturity [48]. Serum levels of haptoglobin were elevated during weaning, which indicates exposure to stress or inflammatory stimuli [49]. In human studies, haptoglobin is considered as an indicator of obesity. It is expressed by adipocytes and its abundance in white adipose tissue and in plasma shows a direct correlation with the degree of adiposity [50,51]. To our knowledge, this is the first study that detects haptoglobin in muscle and adipose tissue in pre-weaning piglets.

Increased levels of n-6 PUFA in the diet may have activated genes related to the glycolysis and gluconeogenesis pathway. *PGK2* encodes the protein that is responsible for the first catalytic ATP-generating step in the glycolytic pathway. Conversely, dietary PUFAs (i.e., LA, EPA, and DHA) are known potent inhibitors of metabolic enzymes and can suppress glycolytic and lipogenic genes [52,53]. Another common protein that was found is liver carboxylesterase. This enzyme is found in the liver and further protects the organ from alcohol- or diet-induced inflammation, damage, or injury [54]. Furthermore, SOY also stimulated an important class of cathelicidins known as protegrins. Cathelicidins are short cationic peptides that are part of the innate immune system for their antimicrobial capacity against Gram-positive and Gram-negative bacteria and their immunomodulatory functions [55,56].

We have also investigated the overabundant proteins and their protein-coding genes in response to SOY dietary treatment that were separately found in either muscle or adipose tissues. The increased detection of MX2, an interferon-stimulated gene induced by Type I interferons in response to viral infections, mainly HIV-1 [57], was only found in piglet muscle tissue. In humans, the PUFA AA has been shown to regulate the binding of interferon in skin fibroblast and n-6 PUFAs have been shown to influence its antiviral function [58,59]. Targeted lipodomics also revealed the correspondence of high levels of AA and low levels of EPA in phospholipid membranes to autoimmune diseases and how dietary supplementation of the n-3 PUFA DHA suppresses autoimmune pathogenesis and blocks gene expression pathways related to interferon [60,61]. Another protein-coding gene that was identified in adipose tissue is DSG1, an adhesive desmosomal protein that preserves human epidermis structure. A study in human squamous cell carcinoma revealed the inhibition of ETA on the expression of desmoglein and how the n-6 PUFA GLA upregulated its activity [62,63]. n-6 also influences the regulation of GSTA2, an enzyme that reduces lipid peroxidation products. Multiple studies reported that AA reduced the activity of GST alpha enzymes in zebrafish and inhibited the hepatic glutathione-S-transferase in mice. On the other hand, DHA activates the glutathione antioxidant systems, preventing oxidative deterioration and associated negative sensory attributes in pork and poultry meat, and it is differentially expressed as a inflammatory response to atherosclerosis in mice [64,65,66]. Muscle and motor protein MYH4 was found to be up-regulated when pig adipose and skeletal muscle were exposed to glucose oxidation-promoting factor, such as cold exposure [67]. PUFA treatments (DHA, EPA, or AA) in muscle cell cultures also significantly decrease myosin heavy chain genes expression levels. By comparing the effect of DHA and AA supplementation on the same study, it could be concluded that n-3 has a greater inhibitory effect on myoblasts [68,69]. The results of the same study showed that DHA and EPA also downregulated MYL4, suggesting that different concentrations of PUFAs can also regulate mRNA expression levels in muscle and adipose tissues.

## 4. Materials and Methods

### 4.1. Experimental Design, Animals, and Housing

The research was conducted at the gestation and lactation facilities of Animal Production Research and Teaching Centre of the Department of Veterinary Medicine and Animal Science, Università degli Studi di Milano (Lodi, Italy). All experimental protocols were approved by the ethical committee of the University of Milan (OPBA 22/2020).

Sows were artificially inseminated with pooled semen according to heat cycle schedule. A total of 8 fourth parity sows (Topig40 × Topigs Fomeva, Topigs Norsvin, Vught, The Netherlands), with an average body weight of 256.56 ± 10.76 kg (mean ± SEM) and body condition score of 3.25 ± 0.89 (mean ± SEM), were selected after pregnancy confirmation by echographic assay performed on d 30 after artificial insemination and were used for the trial. Sows remained at gestation facilities until they were moved to individual farrowing crates on d 108 of gestation. Rooms were equipped with computer-controlled heating and mechanical ventilation systems, and were monitored over a 24 h period. Initial room temperature was set at 28 °C with a ventilation rate of 10 m^3^/h/head, and then gradually decreased by 1 °C/week until to a final temperature of 25 °C at the end of the trial. Sows were then maintained in the farrowing creates until weaning at an average d 25.75 of lactation. The sows were reared on slatted floor with individual feeders and ad libitum access to water.

On newborn piglets, ear notching and tagging, and iron injection were performed within 24 h of birth. Piglets were nourished only with sow’s milk. At the end of lactation, a total of 48 piglets (24 males and 24 females) were selected for fatty acid analysis from a pool of 98 piglets with an average body weight of 7.49 ± 0.16 kg (mean ± SEM). The piglets were further selected down to 24 piglets (12 males and 12 females) for proteomic analysis.

### 4.2. Experimental Diets

Sows were randomly divided between two dietary treatments with n-6/n-3 PUFA ratios of 13:1 (SOY) and 4:1 (LIN) during gestation and lactation. The n-6 and n-3 fatty acid supplementation was derived from soybean oil and linseed oil (Mazzoleni s.pa., Bergamo, Italy) and were both added to the basal diets (Table 5). Sow gestation and lactation basal diets were adjusted to attain final n-6/n-3 PUFA ratios of SOY and LIN, with the addition of soybean and linseed oil, throughout gestation and lactation. All experimental diets were formulated to be isonitrogenous and isocaloric and to meet or exceed the nutrient requirement for gestating and lactating sows [28]. Total fatty acid profile of oil and dietary treatments was presented on Table 6.

Feeding of sows followed the same protocol as described in [7]. Sows were given liquid feed by mixing the basal diets and water. The amount of soybean and linseed oils were calculated and added daily on top to achieve the previously described total n-6/n-3 PUFA ratios of SOY and LIN, for both gestation and lactation feeding plans. The gestational diet was given at 2.5 kg/d with 11 g/d of soybean or linseed oil from the day of insemination to d 59, at 2.7 kg/d with 13 g/d of soybean or linseed oil from d 60 to d 89, and 3 kg/d with 15 g/d of soybean or linseed oil from d 90 until before parturition. Sows were fed per pen (2 sows/pen) with individual feed troughs. The lactation diet was fed at 0.5 kg/d on the day of farrowing (d 0 and then gradually increased to a maximum of 8 kg/d at weaning), with the corresponding amount of soybean and linseed oil to attain the total n-6/n-3 PUFA ratios. Sows were fed twice a day and given ad libitum access to water. Piglets were only nurtured with sow’s milk throughout the entire lactation period.

### 4.3. Recording and Sampling

Body weights (BW) of sows were assessed at the time of insemination, d 30, d 60, before transferring to farrowing crates (d 108), and at the end of lactation. One sow from the SOY group had to be removed from the study due to lameness, which affected its reproductive performance.

Piglets born, born alive, and born dead (stillborn, mummified, crushed, and abnormal) were counted within 24 h postpartum to calculate the survival rate at birth. Piglets were weighed at 24 h postpartum, d 15, and at weaning to calculate average daily weight gain. Feed efficiency cannot be calculated precisely since the piglets were suckling from sows, although ‘weigh–suckle–weigh’ procedure was an option but would have imposed great stress on piglets due to repeated weighing [70]. Litter weights were calculated as the sum of the individual piglet weights per sow per treatment. Piglet and litter weight gains were also calculated and divided into 3 subperiods—(1) At birth (d 0) to d 15, (2) from d 15 to weaning, (3) overall weight gain from birth up to weaning. At the end of lactation, the longissimus dorsi muscle and subcutaneous fat tissues were collected in triplicate during slaughtering, in accordance with European Council Regulation (EC) N° 1099/2009 protocols. The collected samples were labeled and placed in cryovials, snap-frozen in liquid nitrogen, and stored at −80 °C until analyses.

### 4.4. Muscle and Adipose Tissue Samples Fatty Acid Analysis

Extraction and derivatization of fat from muscle and adipose tissue samples were performed according to established lab protocols [71,72]. Fatty acid methyl esters (FAMEs) were measured using ThermoQuest Trace GC 2000 gas chromatography (Thermo Scientific, Bremen, Germany) with a flame ionization detector (Restek, PA, USA); nonadecanoic acid (C19:0; 10 mg/mL of hexane) was the as internal standard and nitrogen (N) was the carrier gas. FAMEs were separated by a fused silica capillary column (Rt-2560, 100 m × 0.25 mm × 0.25 μm) and followed the given program: 70 °C for 5 min; increased by 2 °C $\min^{-1}$ until 240 °C, with total chromatographic runtime of 120 min. Individual FAMEs were verified by comparing peak retention times with standard mixtures (Supelco 37 FAME Mix, Bellefonte, PA, USA) and pure standard methyl esters from Sigma-Aldrich (Saint Louis, MO, USA).

### 4.5. Proteomics Analysis

#### 4.5.1. Protein Extraction, Processing, and Digestion

Approximately 10 mg of longissimus dorsi muscle tissues were homogenized using a Varispeed A581 v.220 Potter homogenizer (Orlando Valentini, Milan, Italy) in 400 μL of extraction buffer (8 M urea, 20 mM Hepes pH 8.0, with Protease inhibitor cocktail) at full speed for 1 min [73]. The homogenate was sonicated using an ultrasonic probe in bursts of 20–30 s and centrifuged at 13,200 rpm for 15 min at 18 °C to pellet the tissue debris. Following dilution of urea to 2 M with NH_4_HCO_3_, 40 μg of protein was reduced (5 mM DTT, 30 min, 55 °C), alkylated (15 mM iodoacetamide, 20 min room temperature in the dark), and digested with (protein:trypsin ratio 40:1, 37 °C, overnight). Digestion was blocked by acidification with 4 μL trifluoroacetic acid. Digested peptides were desalted with Zip-tip according to the manufacturer’s instructions, dried with Speedvac (Thermo Fisher Scientific, MA, United States) and resuspended in 0.1% formic acid (FA) prior to MS analysis [74].

#### 4.5.2. Nano-LC-MS/MS Analysis

Nano-HPLC was performed on a Dionex Ultimate 3000 nano-LC system (Sunnyvale, CA, USA) coupled to an Orbitrap Fusion™ Tribrid™ Mass Spectrometer (Thermo Scientific, Bremen, Germany) equipped with nano-electrospray ion source. Peptide mixtures were loaded onto a Acclaim PepMap 100 − 100 μm × 2 cm C18 (Thermo Scientific) and separated on EASY-Spray column ES802A, 25 cm × 75 μm ID packed with Thermo Scientific Acclaim PepMap RSLC C18, 3 μm, 100 Å using mobile phase A (0.1% formic acid in water) and mobile phase B (0.1% formic acid in acetonitrile 20/80, v/v) at a flow rate 0.300 μL/min. The temperature was set to 35 °C. Samples were injected in duplicate. One blank was run between samples to prevent sample carryover. MS spectra were collected over an m/z range of 375–1500 Da at 120,000 resolutions, operating in the data dependent mode, cycle time 3 s between master scans. HCD MS/MS spectra were acquired on an Orbitrap at resolution of 15,000 using a normalized collision energy of 35%, and an isolation window of 1.6 m/z. Dynamic exclusion was set to 60 s. Rejection of +1 and unassigned charge states were enabled.

#### 4.5.3. Proteomics Data Processing

Thermo raw data were analyzed against a protein database using SEQUEST algorithm in Proteome Discoverer software version 2.5 (Thermo Scientific) for peptide/protein identification [18]. MS/MS spectra were searched against Uniprot KnowledgeBase (KB)/Swiss-Prot *Sus scrofa* database (sp_canonical TaxID = 9823) (v2021-03-31). The minimum peptide length was set to six amino acids and enzymatic digestion with trypsin was selected, with maximum 2 missed cleavages. A precursor mass tolerance of 8 ppm and fragment mass tolerance of 0.02 Da were used; Acetylation (N-Terminus), Met-loss (M) and Met-loss + Acetylation (M) were used as dynamic modifications at protein terminus. Carbamidomethylation (C) was used as a static modification.

A decoy database search was performed to determine the peptide false discovery rate (FDR) with percolator node. The false discovery rates (FDRs) at the protein and peptide level were set to 0.01 for highly confident peptide-spectrum matches and 0.05 for peptide-spectrum matches with moderate confidence. Both unique and razor peptides were selected for protein quantification. Potential contaminants were filtered out using the PD_contaminants_2015 database (# 298 sequences). The abundance ratio for each protein is calculated based on the signal intensity ratios of isotopic pairs of peptide ions detected in the mass spectrometry scans [75].

### 4.6. Statistical Analysis

Data relative to piglet body weight (BW) and fatty acids were analyzed using the general linear model (GLM) procedure in SAS Studio 3.8, on SAS OnDemand for Academics release 9.04.01M6P11072018 (accessed 30 November 2021; https://welcome.oda.sas.com/home; SAS Institute Inc., Cary, NC, USA). Figures were plotted using GraphPad Prism version 8.3.0. (GraphPad software, La Jolla, CA, USA). The statistical model considered the n-6/n-3 PUFA ratios as fixed effects and individual piglets as repeated effect. Least squares means were calculated for each independent variable. The main n-6 polyunsaturated fatty acids, namely linoleic acid (LA), γ-linoleic acid (GLA), dihomo-γ-linolenic acid (DGLA), and arachidonic acid (AA), were grouped and calculated, as well as the major n-3 polyunsaturated fatty acids, such as α-linolenic acid (ALA), eicosatetraenoic acid (ETA), eicosapentaenoic acid (EPA), and decosahexaenoic acid (DHA). Detected saturated fatty acids (SFAs), monounsaturated fatty acids (MUFAs), and polyunsaturated fatty acids (PUFAs) were summed and analyzed. Finally, the total n-6 and n-3 PUFAs and the n-6/n-3 PUFA ratios were summarized. Main effects of diet and tissue, and the interactive effects between the diet and tissue across diets were also determined using Tukey adjustment for multiple comparisons. Data were presented as LSM (least square means) ± SEM (standard error of means) in both tables and figures. The α-level that was used to determine significance was 0.05.

Proteins from muscle and adipose tissues were identified and reported with their designated UNIPROT Accession number. Protein ratios are calculated using Proteome Discoverer software version 2.5 (Thermo Scientific), as the median of all possible pairwise peptide ratios calculated between replicates of all connected peptides using *t*-test pairwise ratio-based approach. Protein quantification was based on the label-free quantification (LFQ) in which the mean LFQ intensities as well as the standard deviation of this value were calculated for SOY and LIN. The fold changes in the level of the proteins were assessed by comparing the mean LFQ intensities among all experimental groups. A protein was considered to be differentially expressed if the difference was statistically significant (*p ≤* 0.05), the minimum fold change was ±2, and a minimum of two peptides were identified.

## 5. Conclusions

The high and low dietary ratios between n-6 and n-3 fatty acids implemented on gestating and lactating sow diets influenced the fatty acid concentrations in both muscle and adipose tissues of pre-weaned piglets, and stimulated proteins and protein-coding genes related to innate immune response and acute inflammatory response. These findings show how low n-6/n-3 PUFA maternal diet can directly affect the early growth performance and fat deposition of suckling piglets. Some evidence of positive changes in immune status of piglets were also outlined by the proteome approach on muscle and fat tissue when sow are fed low n-6/n-3 PUFAs diets, but further studies including liver and serum proteomics could complement these findings, together with a correlation analysis between serum immunological status and proteomic assays.

## Figures and Tables

**Figure 1 molecules-27-02925-f001:**
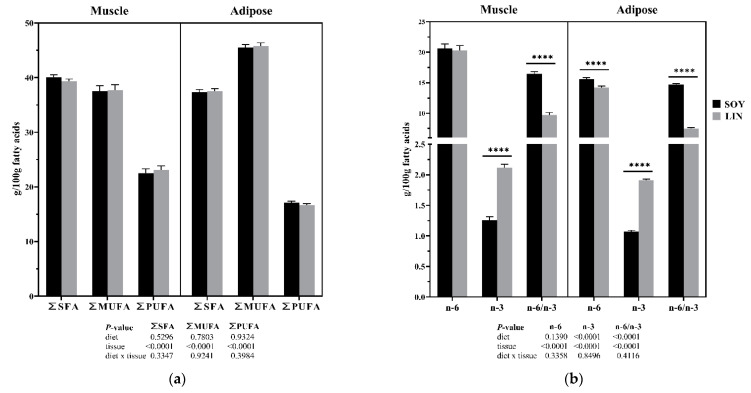
Fatty acid profile of muscle and adipose tissue samples for SOY and LIN diets. (**a**) Sum of saturated fatty acids (SFAs), monounsaturated fatty acids (MUFAs), and polyunsaturated fatty acids (PUFAs); (**b**) total n-6, n-3, and the n-6/n-3 polyunsaturated fatty acid ratio; (**c**) major n-6 polyunsaturated fatty acid profile (linoleic acid, *LA*; gamma γ-linoleic acid, *GLA*; dihomo-gamma γ-linolenic acid, *DGLA*; arachidonic acid, *AA*); (**d**) major n-3 polyunsaturated fatty acid profile (alpha α-linolenic acid, *ALA*; eicosatrienoic acid, *ETA*; eicosapentaenoic acid, *EPA*; decosahexaenoic acid, *DHA*). Data are presented as LSM ± SE; significance levels: * = *p* < 0.05, *** = *p* < 0.001, **** = < 0.0001, no label = not significant, *p* > 0.05; *n* = 24 piglets for each SOY and LIN group.

**Table 1 molecules-27-02925-t001:** Reproductive performance of sows between dietary treatments *.

Item	SOY	LIN	SEM	*p*-Value
**Sow weight, kg**				
insemination	266.00	259.50	12.75	0.73
d 30	271.56	254.50	9.17	0.24
d 60	283.88	269.38	8.51	0.27
farrowing	312.10	296.00	7.98	0.20
**Sow weight gain, kg**				
insemination to d 30	5.56	−4.88	5.31	0.21
d 30 to d 60	12.31	14.88	3.18	0.59
d 60 to farrowing	28.23	26.63	3.38	0.75
insemination to farrowing	46.10	36.63	7.53	0.41
**Number of piglets**				
total born	15.50	14.00	1.31	0.45
born dead	2.50 ^a^	0.50 ^b^	0.41	0.01
weaned	12.00	12.50	1.34	0.80
pre-weaning mortality, %	22.50%	11.00%	0.06	0.19

Data are least square means. Sows were fed diets with dietary treatments of n-6/n-3 polyunsaturated fatty acid ratios of 13:1 (SOY) and 4:1 (LIN). * Eight fourth parity sows; four in each treatment group. ^a,b^ Values within a row lacking a common superscript letter are significantly different (*p* < 0.05).

**Table 2 molecules-27-02925-t002:** Growth performance of pre-weaned piglets *.

Item	*n*	SOY	LIN	SEM	*p*-Value
**Piglet weight, kg**					
d 0	108	1.51	1.44	0.04	0.33
d 15	104	4.75 ^b^	5.30 ^a^	0.16	0.02
at weaning	98	7.52	7.47	0.22	0.89
**Piglet weight gain, kg**					
d 0–15	104	3.19 ^b^	3.85 ^a^	0.14	0.0006
d 15–weaning	98	2.72 ^a^	2.22 ^b^	0.13	0.01
d 0–weaning	98	5.95	6.04	0.20	0.75
**Piglet ADG, kg**					
d 0–15	104	0.21 ^b^	0.26 ^a^	0.01	0.0006
d 15–weaning	98	0.55	0.58	0.02	0.38
d 0–weaning	98	0.23	0.24	0.01	0.52

Data are least square means. Sows were fed diets with dietary treatments with n-6/n-3 polyunsaturated fatty acid ratios of 13:1 (SOY) and 4:1 (LIN). * Number of piglets varies due to accounted mortality before weaning; d 0: SOY 54 piglets, LIN 54 piglets; d 15: SOY 51 piglets, LIN 53 piglets; at weaning: SOY 48 piglets, LIN 50 piglets. ^a,b^ Values within a row lacking a common superscript letter are different (*p* < 0.05).

**Table 3 molecules-27-02925-t003:** Differentially expressed proteins * from muscle tissue, protein-coding genes, and their abundance ratio (SOY/LIN).

UNIPROT Accession	Description	Gene	Abundance Ratio	*P_adj_*-Value
A7VK00	Interferon-induced GTP-binding protein Mx2	*MX2*	100.00	6.50 × 10^−16^
P51524	Prophenin and tritrpticin precursor	-	100.00	6.50 × 10^−16^
Q6RI85	Phosphoglycerate kinase 2	*PGK2*	5.12	1.73 × 10^−8^
Q8SPS7	Haptoglobin	*HP*	2.45	0.01
P02189	Myoglobin	*MB*	2.10	0.06
Q29550	Liver carboxylesterase	*CES1*	2.02	0.09
P32196	Protegrin-3	*NPG3*	2.00	9.55 × 10^−2^

* Out of the 339 proteins identified, only overabundant proteins with P_adj_ value < 0.05 (significant) and <0.10 (tendency) are shown.

**Table 4 molecules-27-02925-t004:** Differentially expressed proteins * from adipose tissue, protein-coding genes, and their abundance ratio (SOY/LIN).

UNIPROT Accession	Description	Gene	Abundance Ratio	*P_adj_*-Value
Q95281	60S ribosomal protein L29	*RPL29*	100.00	1.18 × 10^−15^
Q4PS85	Myozenin-1	*MYOZ1*	100.00	1.18 × 10^−15^
F1RRT2	Myosin light chain 4	*MYL4*	6.93	2.47 × 10^−13^
Q9TV62	Myosin-4	*MYH4*	5.12	3.33 × 10^−9^
Q8SPS7	Haptoglobin	*HP*	3.56	2.66 × 10^−5^
P32195	Protegrin-2	*NPG2*	2.95	7.16 × 10^−4^
Q29550	Liver carboxylesterase	*CES1*	0.51	0.01
Q6RI85	Phosphoglycerate kinase 2	*PGK2*	0.52	0.01
P06348	Histone H1t	*H1-6*	2.37	0.02
P51781	Glutathione S-transferase alpha M14	*GSTA1*	2.34	0.02
Q3BDI7	Desmoglein-1	*DSG1*	0.57	4.96 × 10^−2^
Q29361	60S ribosomal protein L35	*RPL35*	2.15	0.06
P79377	Metallothionein-1D	*MT1D*	2.10	0.08
O77696	Sarcoplasmic/endoplasmic reticulum calcium ATPase 3	*ATP2A3*	2.09	0.08
P79263	Inter-alpha-trypsin inhibitor heavy chain H4	*ITIH4*	2.08	0.09

* Out of the 389 proteins identified, only overabundant proteins with P_adj_ value < 0.05 (significant) and <0.10 (tendency) were shown.

**Table 5 molecules-27-02925-t005:** Composition of basal sow diets.

Item	Gestation	Lactation
**Ingredients (g/kg as fed basis)**		
Barley	200.13	201.93
Maize	150.00	161.00
Wheat	150.00	80.00
Wheat bran	160.00	150.00
Sunflower meal, 36% CP	52.00	50.00
Soybean hulls	50.00	49.00
Soybean meal, 48% CP	40.00	89.00
Maize germ meal	40.00	60.00
Biscuits	31.00	70.00
Animal fat, lard	-	34.60
Fish meal, 64% CP	-	20.00
Calcium carbonate	9.52	16.40
Calcium sulfate	5.00	-
Monocalcium phosphate	3.60	3.70
Vitamin premix *	3.00	3.00
L-Lysine	1.65	3.59
Sodium chloride	1.50	3.50
L-Threonine	1.08	1.31
Magnesium sulfate	1.00	1.00
Attapulgite **	-	1.00
Liquid choline	0.50	0.50
Methionine	0.02	0.37
Tryptophan	-	0.10
**Composition (%DM)**		
Crude protein	14.30	17.00
Crude fat	2.98	6.07
Crude fiber	7.23	6.59
Ash	5.46	5.90
Ca	0.68	0.85
*p*	0.65	0.60
Lysine	0.70	1.07
Methionine	0.25	0.34
Met + Cys	0.53	0.64
Digestible energy (kcal/kg)	2838.16	3200.63

* Providing (per kg of complete diet): vitamin A, 399 × 10^4^ IU; vitamin D3, 650,000 IU; vitamin E, 50,000 mg; vitamin K3, 1355 mg; folic acid, 500 mg; niacinamide 10,000 mg; calcium pantothenate 7500 mg; vitamin B1 1000 mg; vitamin B2 2000 mg; vitamin B6 1000 mg; vitamin B12 10 mg; biotin 333 mg; Fe (as FeSO_4_), 54,974 mg; Cu (as CuSO_4_), 3869 mg; Cu (as copper chelate of hydroxy analogue of methionine), 425 mg; Cu (as cupric chelate of amino acids hydrate), 125 mg; Mn (as MnO), 14,736 mg; Mn (Mn chelate of amino acids hydrate, 250 mg; Zn (as ZnO), 15,287 mg; Zn (as Zn chelate of hydroxyl analogue of methionine), 1085 mg; Zn (as Zn chelate of amino acids hydrate), 615 mg; Se (as Na_2_SeO_3_), 77.9 mg; Selenomethionine from *Saccharomyces cerevisiae* NCYC R646, 22 mg; I (as Ca(IO_3_)_2_), 250 mg; citric acid, 20 mg; orthophosphoric acid, 22.5 mg; butylated hydroxyanisole (BHA), 20 mg; butylhydroxytoluene (BHT), 40 mg; 6-phytase (EC 3.1.3.26), 83,500 OTU; and endo-1,4-beta-xylanase (EC 3.2.1.8), 501,000 EPU. ** A magnesium aluminosilicate mineral clay being used as an absorbent.

**Table 6 molecules-27-02925-t006:** Fatty acids * (g/100 g total fatty acids) of dietary oils and sow diets.

Lipid Name	Common Name	Dietary Oils		Sow Basal Diets		Dietary Treatments			
						Gestation		Lactation	
		Soybean	Linseed	Gestation	Lactation	SOY	LIN	SOY	LIN
C6:0	Caproic acid	*n.d.*	*n.d.*	0.01	*n.d.*	0.01	0.02	*n.d.*	*n.d.*
C10:0	Capric acid	*n.d.*	0.02	0.06	0.09	0.05	0.04	0.06	0.07
C12:0	Lauric acid	*n.d.*	*n.d.*	0.27	0.46	0.23	0.25	0.42	0.41
C14:0	Myristic acid	0.05	0.05	0.41	1.23	0.35	0.36	1.11	1.13
C14:1	Myristoleic acid	*n.d.*	*n.d.*	0.02	0.01	0.02	0.02	0.01	0.01
C15:0	Pentadecylic acid	0.01	0.02	0.08	0.06	0.08	0.07	0.04	0.06
C16:0	Palmitic acid	10.37	5.99	16.72	21.93	15.82	15.2	20.81	20.39
C16:1	Palmitoleic acid	0.06	0.07	0.24	1.2	0.22	0.2	1.09	1.07
C17:0	Margaric acid	0.08	0.05	0.12	0.17	0.12	0.11	0.14	0.16
C17:1	Heptadecenoic acid	0.03	0.03	0.05	0.03	0.04	0.04	0.02	0.01
C18:0	Stearic acid	5.04	3.98	3.84	8.46	4.01	3.86	8.13	8.03
C18:1 n-9 *trans*	Elaidic acid	0.03	0.01	0.08	0.04	0.08	0.07	0.04	0.05
C18:1 n-9 *cis*	Oleic acid	23.29	16.57	24.25	33.49	24.11	23.16	32.5	31.84
C18:2 n-6 *cis*	Linoleic acid	52.46	16.3	49.01	29.58	49.5	44.37	31.8	28.29
C20:0	Arachidic acid	0.41	0.13	0.27	0.23	0.29	0.25	0.25	0.23
C18:3 n-6	Gamma-linoleic acid	0.01	0.22	0.01	0.01	0.01	0.04	0.03	0.04
C20:1 n-9	Eicosenoic acid	0.16	*n.d.*	0.38	0.57	0.35	0.33	0.53	0.51
C18:3 n-3	Alpha-linolenic acid	7.38	56.24	3.36	1.75	3.93	10.86	2.29	7.04
C21:0	Heneicosylic acid	0.02	*n.d.*	0.03	0.02	0.03	0.02	0.03	0.01
C20:2	Eicosadienoic acid	0.02	0.03	0.05	0.26	0.05	0.04	0.24	0.22
C22:0	Behenic acid	0.39	0.1	0.21	0.1	0.23	0.21	0.12	0.1
C22:1 n-9	Erucic acid	*n.d.*	0.03	0.04	0.01	0.03	0.04	0.02	0.03
C20:4 n-3	Eicosatetraenoic acid	0.01	0.05	0.01	0.03	0.01	0.01	0.03	0.04
C20:4 n-6	Arachidonic acid	0.04	*n.d.*	0.03	0.06	0.02	0.03	0.06	0.05
C23:0 n-6	Dihomo-gamma-linoleic acid	*n.d.*	0.01	0.04	*n.d.*	0.03	0.03	*n.d.*	*n.d.*
C22:2	Docosadienoic acid	*n.d.*	*n.d.*	0.04	*n.d.*	0.05	0.04	*n.d.*	*n.d.*
C24:0	Lignoceric acid	0.14	0.09	0.2	0.08	0.2	0.18	0.1	0.08
C20:5 n-3	Eicosapentaenoic acid	0.02	0.01	0.12	0.11	0.1	0.1	0.11	0.1
C24:1 n-9	Nervonic acid	*n.d.*	0.01	0.03	0.02	0.02	0.03	0.02	0.03
Saturated fatty acids (SFAs)		16.5	10.45	22.27	32.83	21.46	20.61	31.22	30.66
Monounsaturated fatty acids (MUFAs)		23.57	16.71	25.09	35.37	24.87	23.89	34.23	33.56
Polyunsaturated Fatty Acids (PUFAs)		59.93	72.84	52.64	31.8	53.67	55.5	34.56	35.78
Omega-3 (n-3)		7.4	56.3	3.48	1.89	4.04	10.98	2.43	7.18
Omega-6 (n-6)		52.5	16.52	49.06	29.64	49.54	44.44	31.89	28.38
n-6/n-3 ratio		7.09	0.29	14.1	15.64	12.27	4.05	13.11	3.95

* *n.d.* = not detected.

## Data Availability

Not applicable.

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
