# Peer review of "Low n-6/n-3 Gestation and Lactation Diets Influence Early Performance, Muscle and Adipose Polyunsaturated Fatty Acid Content and Deposition, and Relative Abundance of Proteins in Suckling Piglets"

_molecules, 2022, doi:10.3390/molecules27092925_

Round 1

Reviewer 1 Report

Abstract

Lines 21-22       Two groups of sows were fed diets with high or low n-6/n-3 polyunsaturated fatty acid ratios of 13:1 (SOY) and 4:1 (LIN) respectively during gestation and lactation.

Line 26:             This conclusion needs to be revised: “ Fatty acid profiles on both muscle and adipose tissues showed great resemblance to the sow diets. As an example, arachidonic acid found in muscle (2 to 2.8 %) and adipose (@ 0.4%) is higher than diets (0.02 to 0.06%).  

INTRODUCTION

Lines 43-46:      As precursors of eicosanoids resulting in antagonistic inflammatory functions (anti- and pro-inflammatory, respectively), n-3 and n-6 may induce substrate competition that can potentially affect metabolic health and inflammatory modulation (3,4).

Line 47:             derived from n-6 PUFAS.

RESULTS

Line 119:           The difference is due to increased relative proportion of n-3 PUFAs in the LIN diet expressed as %. The present article is not reporting concentrations in mg FA per g of tissue.

Line 127:           This paper indicates that supplementing sow with a n-6/n-3 ratio during gestation and lactation did not affect the content of the major n-6 PUFAs in the muscle of the piglet. Not knowing the concentrations of fatty acid, expressed in mg/g of tissue, it is thus appropriate to say that these diets did not affect the relative proportion of n6/n3. The muscular content in n-3 and n6 is not reported.

METHODS

Line 404:           This table shows the distribution of fatty acids expressed as %. What is the amount in mg of n-3 and n-6 fatty acids per g of total diet ? Was the amount of diet given and ingested by both groups SOY and LIN equal ?

Based on % distribution (table 6), the n6/n3 ratios of soy and lin dietary oils were respectively 7.09 and 0.29 whereas sow basal gestation diet had a ratio of 14.1. How much oil was thus needed to decrease the basal diet ratio from 14.1 to obtain a gestation SOY ratio of 12.27 ? AND to obtain a gestation LIN ratio of 4.05 ? Knowing that SOY and LIN diets were isocaloric, were the total amount of fatty acids added to basal diets similar? Data showing the amount of fatty acids expressed in mg per g of diet and the resulting ratios are thus needed. Similarly, how much oil and which oil is required to decrease the basal 15.64 ratio (basal lactation diet) to 13.11 (SOY treatment) and 3.95 (LIN treatment)?

Line 494:           (…) the major n-3 polyunsaturated fatty acids such as α-linolenic acid (ALA), eicosatrienoic acid (ETA), eicosapentaenoic acid (EPA), and decosahexaenoic acid (DHA).
DHA (C22:6, n-3) values are not reported in Table 6. Furthermore, summation of each reported fatty acids of the dietary oils and dietary treatments, with no DHA, equal 100%. However, fig 1 shows that the LIN diet can significantly increase DHA levels in both muscle and adipose tissues. Knowing the absence of fatty acid elongation in this experimental model and the absence of DHA in the diets, can this observation be explained?

DISCUSSION

Line 239:           It known that fatty acid content of adipose tissues is in the order of 90% of its cellular mass whereas fatty content of muscular cell is markedly lower. The authors are reporting that “ The amount of fatty acids found in muscles are significantly higher in quantity than that of adipose, (…). The present paper is not reporting the concentration or the content of fatty acid in adipose tissue or muscle but rather the fatty acid distribution.

CONCLUSION

Line 522:           The aim of this paper was to show correlations between dietary fatty acid composition and tissue fatty acid profiles, as well as comparisons between protein composition in animals receiving both diets. Thus, this paper is not designed to explain mechanisms as suggested by the authors as indicated in the following sentence: “ These findings (…) provide clear illustration on how maternal diet and its milk directly affects the early growth performance, fat deposition, and immune status of suckling piglets, (…).”

General comments: This paper is an observational study. Fatty acid data confirm previously published reports. Protein data are original observations and are used to generate hypotheses on metabolic pathways.
Fatty acid data need to be revised taking into consideration diet and tissue fatty acid concentrations expressed in mg fatty acid per g of tissue as discussed by Sergeant et al, British J Nutrition (2016) 115, 251-61.

Author Response

Dear reviewer 1, thank you for your comments and suggestions. Please see the attachment for the answers and clarifications. For your perusal.

Reviewer 2 Report

Dear Authors,

The title is a little bit exaggerated, but the study is interesting. However for me it is only a preliminary study because of small number of sows included them. Only 4 sows in a one group taking into account individual variability, makes it much more difficult to conclude correctly. In my opinion this experiment should be repeated three times to obtain more reliable results for comparisons / calculations and conclusions.

After studying this manuscript, I have some doubts, answer me please:

- Were the results described in this manuscript obtained during the study presented in:  "Nguyen, T. X.; Agazzi, A.; Comi, M.; Bontempo, V.; Guido, I.; Panseri, S.; Sauerwein, H.; Eckersall, P. D.; Burchmore, R.; Savoini, 565 G. Effects of Low Ω6:Ω3 Ratio in Sow Diet and Seaweed Supplement in Piglet Diet on Performance, Colostrum and Milk Fatty 566 Acid Profiles, and Oxidative Status. Animals 2020, 10 (11), 2049 "- I expact that not.

Why am I answering? - because I can't find in reviewed manuscript  information about composition of sows colostrum and milk while in key words it exists. I'm suprised

- there is no sows' body weight on weaning day (Tab.1)  so you can't assess loss during lactation period (line 81).

- sows were at the same parity, I hope that in the same body condition (you should add this information). That's a pity you didn't measure sows' backfat (at begin of the experiment, on the delivery day, 7th day of lactation - if it was a day of milk sampling and on weaning day). It could be useful for assessing excessive uncontrolled lipomobilization during a negative energy balance after delivery in sows from both groups and you could use it to explain the differences in fat content in milk (if it was measured) and of course to compare it with results obtained from piglets' tissues.

- to say the true I don't see any results which could be useful for assessing the immune status of piglets. I agree that probably it is the first study that detects the haptoglobin concentration in muscle and adipose tissue - but because of it you can't assess the immunity status of piglets. To do it you should have measured Hp concentration in piglets blood and later you can compare it with your first results.

"Stimulation of proteins and protein-coding genes related to innate immune response and acute inflammatory response possibly correlated with the pro-inflammatory function of n-6  PUFAs. Although, these findings can bridge the lack of data of the pre-weaning stage of piglets and provide clear illustration on how maternal diet and its milk directly affects the early growth performance, fat deposition, and immune status of suckling piglets, using a proteomics approach."   

I would be careful in drawing such conclusions, we have too little data here on the immune status of piglets. I agree with these sentences in discuss but to use it in conclusion you should measure additionally some immunological parameters in piglets blood at least on a slaughtering day, perhaps it will be enough just total Ig or IgG to confirm such expectations here.

It is my general comments, I understand that we can't do/measure everything what could be useful for us during different experiments, but correct what is possible in this manuscript. First change the title, introduce some changes and good luck in your work.  It is interesting idea, we have some news here.

Author Response

Dear reviewer 2, thank you for your comments and suggestions. Please see the attachment for the answers and clarifications. For your perusal.

Reviewer 3 Report

The work is interesting, is related to the Journal scope, and it was well written. The introduction introduces the subject but the methodology requires a slight explanations. The results were clearly described and presented, in the discussion the results were correctly interpreted and correct conclusions were drawn.

Below are some minor comments:

Please indicate the breed of pigs.

Body weight need not be reported with this accuracy.

If there were 8 sows, fed 2 pcs per pen, the statistic n = 4 is a bit too little. Has a calculation been made regarding the minimum number of animals?

Please provide information on animal fat (Table 5) in the diet for sows. What was that fat?

Were the piglets weaned at 25 days of age?

Please add information on how the animals were sacrificed.

l.428 on which analyzer; please provide the full name and manufacturer.

l.440 please provide the full name.

Author Response

Dear reviewer 3, thank you for your comments and suggestions. Please see the attachment for the answers and clarifications. For your perusal.

Round 2

Reviewer 1 Report

Discussion

Line 494 (now lines 508-510)

In your comments, you correctly indicate that DHA could be synthesized through an alternate EPA-DPA-TPA-THA pathway. Knowing that EPA levels (0.1%) are identical in each dietary treatment and that there is no DHA present in the diets (Table 6), how can the treatment explain the increase in EPA and DHA found in the LIN compared to SOY groups (Fig 1d) ? A comment would be appropriate in the discussion section.

Discussion

Line 239 (now line 250)

Thank you for replacing the terms concentrations by relative proportion throughout the result section. Consequently, the following comment in the discussion section needs to be also adjusted.

 “ The amount of fatty acids found in muscles are significantly higher in quantity than that of adipose, whereas, the differences among fatty acids within adipose tissues between dietary treatments were more defined. »

This observation needs to be supported by appropriate reference to the current literature or data from your paper. You are referring to the AMOUNT of fatty acid, meaning concentrations of fatty acids by gram of tissue. However, your data are showing relative proportions in %. A relatively high percentage (proportion) of a specific fatty acid in a tissue could also represent an extremely low concentration (amount) of fat in the same tissue. In general, it is known that the amount of fat in adipose tissues is higher than muscles.

Later in the same paragraph, the authors report that: “ Although contrary to our results, studies have shown that the fatty acid content in adipose tissue should be much higher than muscle, but the composition is generally similar.” Your current data do not report on the quantity of fat but rather on the relative percentage of each fatty acid. Thus, when the % of one fatty acid increases the % of one or more other fatty acids must decrease to keep the total distribution at 100% in each tissue. Are the values in micrograms of fatty acid per mg of fat tissue or muscle available in the current study?

You emphasize the importance of increased concentrations of n-3 and n-6 fatty acid on the production of pro- or anti-inflammatory eicosanoids. Even though the relative proportion (%) of a specific fatty acid compared to other fatty acids may increase or decrease, its absolute concentration in mg of tissue may remain unchanged. Thus, one cannot conclude that variations in proportions generate similar variations in tissue concentrations. Please note that the amount of eicosanoids generated from fatty acids varies based on their  tissue concentrations not based on their relative distribution in these tissues.

Author Response

Hello Reviewer 1, thank you for all the comments and insights. See attached file for our answers. Have a good day!

Reviewer 2 Report

Dear Authors,

thank you for answers, I have no more comments

Author Response

Hello Reviewer 2, thank you for all the comments and insights. We appreciated them. Have a nice day!